

# An efficient transfer learning based cross model classification (TLBCM) technique for the prediction of breast cancer

Sudha Prathyusha Jakkaladiki and Filip Maly

Department of Informatics and Quantitative Methods, Faculty of Informatics and Management, University of Hradec Králové, Hradec Králové, Czech Republic

## ABSTRACT

Breast cancer has been the most life-threatening disease in women in the last few decades. The high mortality rate among women is due to breast cancer because of less awareness and a minimum number of medical facilities to detect the disease in the early stages. In the recent era, the situation has changed with the help of many technological advancements and medical equipment to observe breast cancer development. The machine learning technique supports vector machines (SVM), logistic regression, and random forests have been used to analyze the images of cancer cells on different data sets. Although the particular technique has performed better on the smaller data set, accuracy still needs to catch up in most of the data, which needs to be fairer to apply in the real-time medical environment. In the proposed research, state-of-the-art deep learning techniques, such as transfer learning, based cross model classification (TLBCM), convolution neural network (CNN) and transfer learning, residual network (ResNet), and Densenet proposed for efficient prediction of breast cancer with the minimized error rating. The convolution neural network and transfer learning are the most prominent techniques for predicting the main features in the data set. The sensitive data is protected using a cyber-physical system (CPS) while using the images virtually over the network. CPS act as a virtual connection between human and networks. While the data is transferred in the network, it must monitor using CPS. The ResNet changes the data on many layers without compromising the minimum error rate. The DenseNet conciliates the problem of vanishing gradient issues. The experiment is carried out on the data sets Breast Cancer Wisconsin (Diagnostic) and Breast Cancer Histopathological Dataset (BreakHis). The convolution neural network and the transfer learning have achieved a validation accuracy of 98.3%. The results of these proposed methods show the highest classification rate between the benign and the malignant data. The proposed method improves the efficiency and speed of classification, which is more convenient for discovering breast cancer in earlier stages than the previously proposed methodologies.

Corresponding author
Sudha Prathyusha Jakkaladiki,
sudha.jakkaladiki@uhk.cz

## INTRODUCTION

In the modern world, although there are much medical equipment and advancements in medicine, nearly two million women are affected by the cruelest disease, breast cancer. Cancer occurs in the human body when there is a cell growth called mutations. Few prominent techniques and methods are available to predict breast cancer in its early stages in middle-aged women. In particular, the mutation divides the cell and grows unconditionally in a chaotic way. Moreover, the progress of this mutation will be abnormal. Consequently, it ends up in the formation of tumor cells in the human body. Breast cancer occurs particularly in a female body when there is a malignancy called a cancerous cell. As it grows continuously, it may spread to other parts of the human body and could automatically result in death. Early detection of the disease will help the women to survive and decrease the death rate. The classification of mammogram images to detect breast cancer is based on certain techniques such as fuzzy systems, machine learning, and deep learning. Two key obstacles prevent the widespread use of AI-based approaches in healthcare settings. Specifically, ML and DL models cannot generalize well in real-world, complex medical datasets. Constraints on data sharing due to ethical and legal considerations, as well as the limited availability of labeled high-quality datasets, all play a role in the creation of new medicines. Image processing converts the data obtained from medical devices such as mammograms. Also, they are converted into a digital form of processing. During the conversion of images to digital format, some useful information would be extracted from the images for analyzing the data. Exactly, the pixels are the information about a particular value in a precise location. The image processing is performed through the following steps.

1. The first is the image obtained from medical devices such as optical scanners or X-rays.
2. Enhancement of the images, including data compression to extract the features.
3. Result will be obtained by analyzing the quality and the classification.

The classification of mammogram images is done by using supervised or unsupervised machine learning algorithms. These algorithms are based on the shape of the cancer cell, which is gentle, malignant, or abnormal.

The goal of the ongoing studies in this field is to suggest a tumor result for breast cancer screening that is more decisive and can be agreed upon by a large number of people. In a similar spirit, getting through the newly erected obstacles brought on by the drawn-out screening process, especially when it requires the use of a photograph, is essential. Mographic imaging as a science and its potential applications have been revitalized by recent advances. Thermography is used for many different purposes, but one of the most promising is breast cancer screening. Nevertheless, thermography has not yet been verified as the preferred technology for doing this particular task.

Computer-aided design detection (CAD) on mammogram images is used to detect the occurrence of breast cancer (*Rao et al., 2010*). Although the performance of the techniques is better, it has many disadvantages as well. Specifically, handcrafted feature extraction is the most tedious process. Compared to the other machine learning technologies, the

traditional method involves a lot of hand-crafted features. It is very difficult and has no generalized procedure. The solution for the existing traditional handcraft-based feature extraction is the convolutional neural network (*Lo et al., 1995*).

According to previous research, the convolutional neural network performs better than the traditional method for getting features from the images. The classic convolutional neural network alexnet has won the challenge in imagenet, where it contains 1,000 classes of colored images and it has got 83.6%. The alexnet is trained on a huge amount of data, which is nearly 1.2 million image databases. Although it works on this kind of database, it is not efficient for medical images, where feature extraction is very difficult to perform. The best way of handling these kinds of medical images is the usage of pre-trained models. The model is trained exclusively over a large number of images and datasets and can be used on a smaller dataset or it can be retrained for the prediction and feature extraction. Therefore, this technique is called transfer learning, which has a greater impact on computer vision-related fields and research (*Tajbakhsh et al., 2016*).

In this proposed research work, the cross-modal deep learning approach is used to predict breast cancer from mammographic images. Preprocessing of images and classification is a part of the model. The images are preprocessed for enhancement and extraction of important information from the images. Moreover, the image is resized for clarity so as to fit into the model. The model includes CNN, transfer learning, ResNet, and Descent.

The key contribution of our research work is as follows:

1. The features are extracted using the convolutional neural network along with the transfer learning technique.
2. The weight and the biases of the pre-trained model are fine-tuned automatically to analyze the features of the mammogram images.
3. ResNet and DenseNet models are employed to compare the accuracy.
4. The learning rate change and data augmentation apply to avoid the over-fitting of the model.

The remaining parts of the article are arranged in the following order. In Section 2, the literature reviews on convolutional neural network transfer learning and augmentation on the mammographic image dataset are analyzed. In Section 3, the dataset and the experimental setup are discussed. In Section 4, the results of the proposed model are discussed. In Section 5, the conclusion is given.

## RELATED WORK

Deep learning is a concept that has been established to extract the pertinent information effectively from the raw images and use it for the classification process in order to overcome the limitations of classic machine learning approaches (*LeCun, Bengio & Hinton, 2015*; *Bengio, Courville & Vincent, 2013*).

The convolutional neural network can be applied in three major ways: (1) pre-trained model of CNN; (2) training the CNN from the scratch; (3) fine-tuning of CNN. *Ganesan*

*et al. (2012)*, have shown numerous machine learning algorithms that are employed for detecting breast cancer on mammographic images. In the study, the databases that are the most commonly used are Digital Database for Screening Mammography (DDSM) and MIAS, as proposed by *Pub et al. (2000)*. They have used a 10-fold cross-validation technique to test the model on the database. *Khan et al. (2017)* and *Zhang et al. (2016)* used the automatic feature extraction method with CPS instead of the handcraft feature extraction technique such as fractional Fourier transform and Gabor filter along with the classifiers such as Support Vector Machine (SVM).

*ElOuassif et al. (2021)* has proposed many neural networks and example methods for the classification of breast cancer. *Nassif et al. (2022)* had focused on convolutional neural networks. It produced excellent results when compared to the other neural networks in terms of feature extraction and classification in mammogram images. *Hussain et al. (2017)* had proved in his study that the convolutional neural network was superior to the other traditional machine learning techniques for extracting the features along with transfer learning using CPS. Deep learning techniques are implemented in the medical field for obtaining better outcomes, along with tough challenges in the input data compared to the other fields. Transfer learning is used by different physicians and technicians for analyzing various medical images, which gives better results in diagnosis (*Litjens et al., 2017*; *Shen, Wu & Suk, 2017*).

*Hussain et al. (2017)*, had achieved 88% accuracy in predicting breast cancer based on the data in the Digital Mammography Training Database (DDSM) which was the smallest data set. He used a pre-trained model, transfer learning, for analyzing the dataset. VGC-16 was employed in his research work as a transfer learning for the identification of breast abnormality in mammogram images with the help of CNN. The VGC-16 is the pre-trained convolutional neural network, which is the best computer vision model. *Alkhaleefah et al. (2020)* enhanced the study accuracy of breast cancer by utilizing the method of double shot transfer learning (DSTL) with the help of pre-trained networks. Unlike the other models which use the smallest medical data, the proposed model employed a larger data set as a target data set for fine-tuning. The weight and the bias were fine-tuned from the pre-trained model. The fine-tuning of parameters was achieved with the help of the target dataset.

An approach for deep learning (DL) had been proposed by *Sun et al. (2017)* dubbed D-SVM for human breast cancer prediction prognosis. Further, the program had successfully discovered order and created an abstract representation from the raw input data which was a combined approach of traditional classification (*Sun et al., 2017*). In order to identify the architectural distortion from digital mammography, *Oyelade & Ezugwu (2020)* examined various deep-learning techniques. The primary objective of their research was to find the abnormalities that are the signs of advanced diseases like masses and micro-calcification. Only 12% of the existing literature, according to their investigation, had used survey data in computer vision and deep learning to create superior computational models for the identification of architectural distortion. In addition to that, it was found that Gabor Filters were used in roughly 78% of the existing literature.

**Table 1 Comparative table of research work.**

| Author | Proposed methodology |
| --- | --- |
| Ganesan et al. (2012) | Machine learning algorithm |
| Zhang et al. (2016) | Fractional Fourier transform and Gabor filter with SVM |
| ElOuassif et al. (2021) | Neural networks |
| Nassif et al. (2022) | Convolutional neural networks |
| Hussain et al. (2017) | CNN and transfer learning using CPS |
| Hussain et al. (2017) | Transfer learning, VGC-16 |
| Alkhaleefah et al. (2020) | Double Shot Transfer Learning (DSTL) |
| Perre, Alexandre & Freire (2018) | VGG, Caff and VGG-m |
| Spanhol et al. (2015) | Speed up Robust Features (SURFs) with SVM |
| Gopal et al. (2021) | Convolutional neural networks with slide images |
| Sun et al. (2017) | IOT and machine learning |

Perre, Alexandre & Freire (2018) proposed a transfer learning approach with the help of three kinds of networks; (1) VGG-f, (2) Caffe, and (3) VGG-m. Particularly, the output was tested in two cases while fine-tuning the model. In the first case, image normalization was applied to find the abnormal letters in mammogram images. In the second case, the model of no image normalization was applied for testing. In the proposed model, both support vector machines and convolution neural networks are employed. Features are extracted using the convolution neural network, and the classification is handled by the support vector machine. The extracted features by the convolution neural network are passed to the support vector machine for the classification of mammogram images for identifying the possibilities of breast cancer.

The study is aimed to develop an end-to-end deep learning framework for multi-label breast lesion detection, which is motivated by the success of CNN's in single-label mammography classification (Karahaliou et al., 2008; Andreadis, Spyrou & Nikita, 2014; Li et al., 2017). In order to create an autonomous multi-labeling framework that aids the radiologist in providing his patients with a thorough report and more accurate diagnoses, it is intended to make use of highly expressive CNN (LeCun, Kavukcuoglu & Farabet, 2010; Mahrooghy et al., 2015; Ekici & Jawzal, 2020; Wang et al., 2020; Shu et al., 2020). Numerous microwave imaging techniques, including microwave tomography and radar-based imaging, were researched. Some imaging diagnostic techniques were described for the diagnosis of breast cancer. It looked into how computer vision and machine learning could detect breast cancer. On the basis of mammographic images, the effectiveness of various approaches was examined. Table 1 shows the related research comparison.

The 3D-based classification for the breast cancer dataset is efficiently performed using attention-based convolutional neural networks (Huang et al., 2022). Cloud computing is used as a resource for storage and processing. Further enhanced CNN is widely used in image classification approaches in major disciplines like medicine, forensics, etc. (Zhou, Wang & Wan, 2022; Zeng et al., 2020). Due to advancements in 3D applications, recent deep learning like a multimodel classifier with cross-correlation is used for image

classification (*Liu et al., 2022b*). The endoscopic images are shaded with affected regions based on the stereo vision algorithm (*Cao et al., 2022*). The space applications images are processed using multimodal medical image processing techniques (*Ban et al., 2022*). Deep learning algorithms like improved pulse-coupled CNN, and transfer learning are used in challenging image processing applications (*Liu et al., 2022a*; *Zhao et al., 2022*). Unsupervised deep learning models in health, business, and e-commerce provides various decision process based on image processing datasets (*Xie et al., 2022*; *Zhang et al., 2021*). Computer vision is not limited, today every digital application requires vision technologies for biometrics and other identification applications (*Zeng et al., 2020*).

*Kaur, Singh & Kaur (2019)* proposed a new feature extraction technique, Speeded-Up Robust Features (SURFs), which is k-means convergence on a mini dataset. An additional layer was introduced in the classification process. 70% of the data from the dataset was used for training and 30% of the data was used for testing models. Classification support vector machines and deep neural networks were employed. The result obtained from the proposed model proved that the deep neural network outperformed better than the decision tree model. The accuracy was improved by employing the convolution neural network model on numerous datasets, which was equal to 87.5%.

This article (*Vulli et al., 2022*) presented a unique technique Densenet-169 for automatic metastasis diagnosis and identification using entire slide pictures, based on the Fast AI framework and the 1-cycle strategy. It also contrasted the specific procedure with the others that had been used in the past. With an accuracy of over 97.4%, the suggested model outperformed well than the previous state-of-the-art approaches.

For accurate classification of breast cancer to forecast whether it is malignant or benign, the proposed research has presented the approach TLBCM, which combines ResNet, Dense net, and CNN network. In specific, ResNet and DenseNet transfer learning are used to learn about the features of the dataset. The experiment has demonstrated the usefulness of information transfer between fields that may at first glance seem unrelated. It is suggested to begin the learning process with an initialization that is based on previously taught weights. The network must be fine-tuned for the new dataset by gradually modifying the loaded weights. The layers with the slowest learning rate have their backpropagation reset in the specified process.

Deep learning techniques built on Convolution Neuronal Networks (CNN) have recent considerable success in the field of biomedical image analysis. Examples include the recognition of mitotic cells in microscopic pictures (*Malon & Cosatto, 2013*), the detection of tumors (*Cruz-Roa et al., 2014*), the segmentation of neural membranes (*Cruz-Roa et al., 2013*), the detection and classification of immune cells (*Ciresan et al., 2012*), and the quantification of mass in mammograms (*Esteva, Kuprel & Thrun, 2015*; *Chen & Chefd'Hotel, 2014*; *Dhungel, Carneiro & Bradley, 2015*).

## PROPOSED TLBCM ARCHITECTURE

The best cancer detection method is shown in Fig. 1. The deep learning approaches CNN, ResNet, and DenseNet are used to extract the salient features from the mammogram image dataset. The dataset is processed to identify and classify the images based on benign and

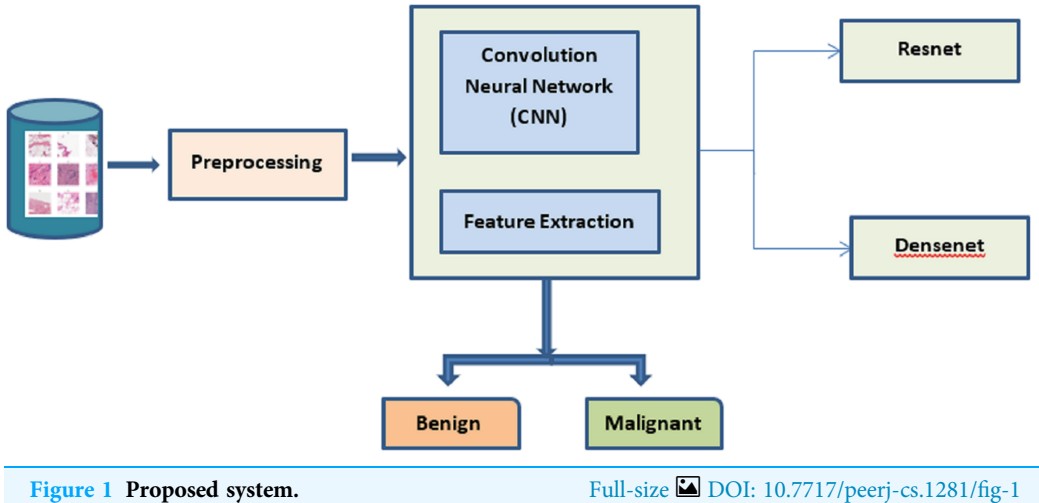

**Figure 1 Proposed system.**               

malignant. The TLBCM method, which includes ResNet, Dense net, and CNN network, is presented in the proposed research as a method that may accurately classify breast cancer in order to predict whether or not it is a malignant form of the disease. Transfer learning methods like ResNet and DenseNet are utilized here in order to acquire knowledge of the characteristics of the dataset. It is recommended that the process of learning start with an initialization that is determined by weights that have been learned in the past. For the newly collected data, the network has to have its loaded weights gradually adjusted in order to achieve optimal performance. During this procedure, the backpropagation of the layers that have the slowest learning rate is reset.

The trained ResNet and DenseNet are used for the image dataset because of the following reason. ResNets, in particular, are a type of neural network architecture that is considered to be among the most effective due to their ability to assist in preserving a low error rate even farther into the network. Convolution is utilized by DenseNet in order to bring the dimension of the input feature maps of each layer down to a constant multiple of the growth rate. The learning rate is set to be adjusted for each iteration based on the training of data.

## Preprocessing

In this section, the pre-processing of data is described. Data augmentation is applied to the data set to increase the number of data to avoid over-fitting during the training and testing of the model. Moreover, the main issue with the medical dataset is the minimum number of samples. The particular dataset can be enlarged with the help of data augmentation and scaling. The missing data cells are replaced with zero. The benign and the malignant data are balanced by introducing a random noise in the data. The 241 malignant data are doubled as 482. In fact, it is achieved by the array of random numbers. The standard deviation sigma value is 0.1. The feature values are scaled using the formula

$$\frac{v_f - v_{min}}{v_{max} - v_{min}} \tag{1}$$

$v_f$ is the feature value. $v_{min}$ is the minimum value, $v_{max}$ is the maximum value.
The square root transformation is applied to the resultant dataset.

## Convolution neural network with transfer learning

Transfer learning is the concept of taking the already learned features from other works to apply to the current problem without the need to start from the beginning. Normally, transfer learning is built on CNN on a well-known dataset. The convolution neural network reduces the input and analyzes the features to differentiate it from the other images. It contains several layers that include a convolution layer, max pooling layer, fully connected layer, and batch normalization. The convolution layer in the CNN extracts the features from the given input mammogram images. In this layer, the images are convolved with a kernel or filter. The kernel is multiplied by the match patch. Further, the filter size is matched with the input size and width, and height of the filter according to the network deployment. After the convolution, the input is transformed through the subsampling which can be max pooling, min pooling, or average pooling. Moreover, the pooling filter is chosen as an odd number. It is responsible for dimensionality reduction which results in minimizing the overfitting issues. The max-pooling layer in the model is used to minimize the sample in the proposed model. Therefore, it is used to reduce the complexity of the model when the data is transferred from one layer to another layer. Additionally, it is used to initiate invariance and the fully connected players on the top of the model are connected to each other.

The proposed architecture first proposed the idea of residual blocks as a means of addressing the issue of vanishing and bursting gradients. In this particular network, the advantage of a method known as skipping connections is made. Activations on one layer can be connected to those on subsequent levels *via* the skip connection, which bypasses those layers in between. Moreover, the results in a residual block are being formed. The formation of resents involves the successive stacking of these residual blocks.

In the proposed method, ResNet 50 is used which is inherited from the base model ResNet. Figure 2 shows the architecture of ResNet 50. In the same way, it comprises nearly 50 layers with the residual block. Further, it reduces the computation process and the complexity of the model. ResNet 50 comprises convolution layers, normalization layers, and residual blocks. There are 16 residual block modules in between the pooling layers. The filter size used here is 3 × 3. This pointer is used to perform the spatial convolution operation for the classification of benign and malignant images in the dataset. The input value $x$ is mapped to the features of output $O(x)$.

$$O(x) = F(x) + x \tag{2}$$
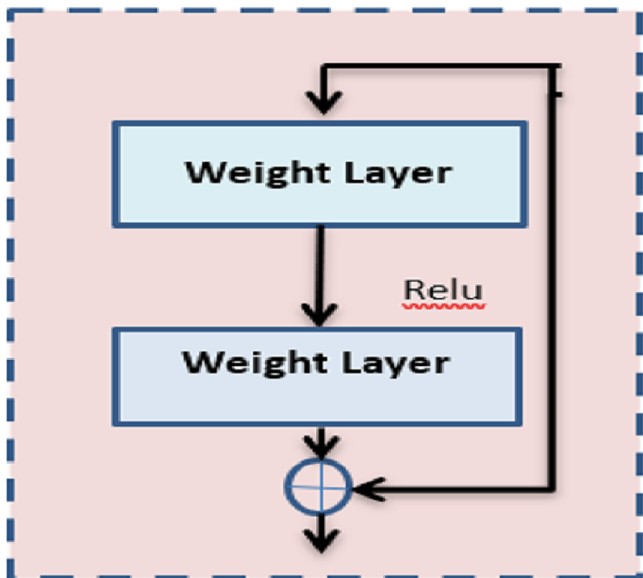

**Figure 2 ResNet architecture.**

The error from the output is $F(x)$. If the value of $F(x)$ is 0, the output feature is exactly the same as the target output. Otherwise, it deviates from the target and the weight needs to be adjusted in the hidden layers.

$$y = F(x, \{V_j\}) + V_j x \qquad (3)$$

$V_j$ is the parameter value that implies the weight to be adjusted in the input shape.

DenseNet is the inherited idea of ResNet. The DenseNet does not contain any constraints for the number of convolution layers and the width of the layers. The parameters are reduced in the DenseNet; hence it avoids redundant features with the help of the feature-reused method. DenseNet is used to avoid the vanishing gradient problem. The dense net concatenates the feature map from different layers to reuse the learned features from the previous layers. The parameters are reduced in the DenseNet; hence it avoids redundant features with the help of the feature-reused method. DenseNet is used to avoid the vanishing gradient problem. The dense net concatenates the feature map from the different layers to reuse the learned features from the previous layers. Let $D_i$ is the output of the $i^{th}$ layer.

$$D_i = H(D_{i-1}) \qquad (4)$$

Then the dense connection of the layers is termed as,

$$D_i = H(D_0, D_1, D_2 \ldots \ldots, D_{i-1}) \qquad (5)$$

The entire architecture of TLBCM architecture is shown in Fig. 3. The pre-trained model can be utilized in order to extract the features, and then those features are sent to the fully connected layer, where they are employed for the categorization of cancer in order to

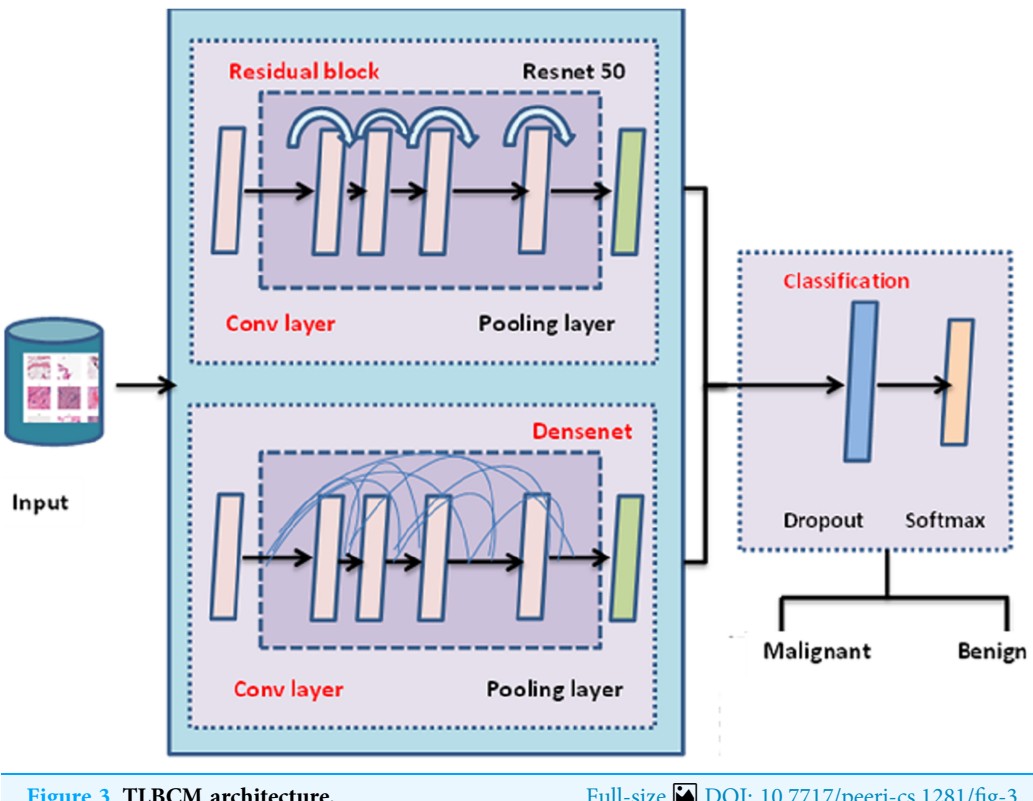

**Figure 3** **TLBCM architecture.**

determine whether or not it is malignant. The transfer learning is done with the help of ResNet 50 architecture, which is used as the pre-trained model to reduce the time complexity and space complexity. In fact, DenseNet uses the features extracted from the previous layer for the training. The dropout value of the model is 0.5.

Table 2 Shows the CNN architecture which is integrated with the pre-trained model. The convolution layer and the fully connected layers are associated with the activation function (Rectified Linear Unit) ReLU. The output layer is associated with the Softmax activation function which is used to distinguish the malignant and benign data.

## EXPERIMENTAL SETUP

The experiment is carried out on the Python 3.6 environment, Keras as a framework and Tensor Flow as a backend. Especially, two datasets are employed for the proposed research work; Breast Cancer Wisconsin (Diagnostic) and Breast Cancer Histopathological Database (BreakHis). Further, there are several datasets pertaining to breast cancer that may be utilized for the development of computer-aided diagnostic systems (CADs) utilizing either deep learning or conventional modeling techniques. On the other hand, the majority of these datasets force practitioners to make a variety of trade-offs, either in terms of their availability or their inherent therapeutic usefulness. In order to get around these restrictions, public datasets known as BreakHis and Wisconsin have only recently been made available.

**Table 2 Table CNN training architecture integrated with transfer learning.**

| Layer | Particulars |
|---|---|
| Input | RGB image |
| Convolution layer 1 | Conv_3-32 + ReLU |
| Max pooling layer 1 | MaxPool_2 |
| Convolution layer 2 | Conv_3-32 + ReLU |
| Max pooling layer 2 | MaxPool_2 |
| Convolution layer 3 | Conv_3-64+ ReLU |
| Max pooling layer 3 | MaxPool_2 |
| Fully connected layer | FC_64 + ReLU (with dropout = 0.5) |
| Output | (Softmax) |

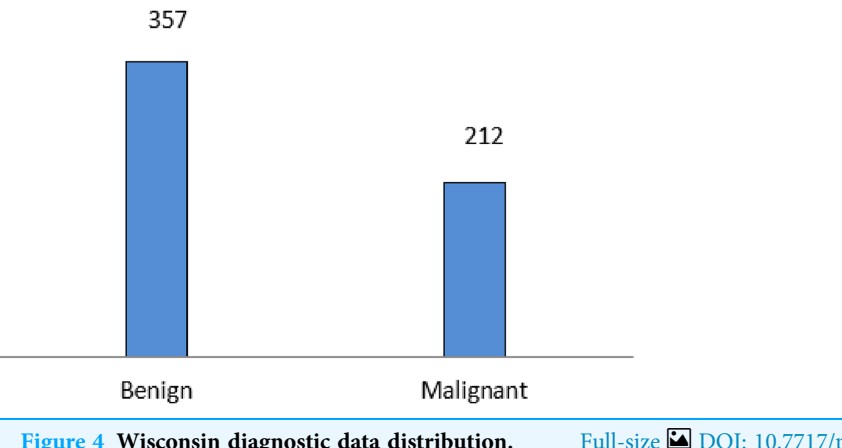

**Figure 4 Wisconsin diagnostic data distribution.**

## Dataset

The experiment is carried out on the Breast Cancer Wisconsin (Diagnostic) data set, a multivariate dataset that comprises 569 instances with 32 attributes. The dataset comprises 357 malignant and 212 benign data. Figure 4 Shows the data distribution of the Wisconsin database. The dataset comprises 357 malignant and 212 benign data.

Figure 5 shows the dataset features' mean value. The mean values are calculated in terms of benign and malignant. It shows the linear pattern between the area, perimeter, and radius.

The dataset comprises ten real-valued features for each cell nucleus. Table 3 shows the real-valued features and their mean values for sample data. It includes radius, texture, area, perimeter, compactness, smoothness, concave points, concavity, fractal dimensions, and symmetry.

The malignant cancer cells are the breast tumors and the benign cell structures are the non-cancer and normal cell structures. The benign are the lumps that are not identified as cancer cells are shown Fig. 6.

Another dataset that is used in the proposed article is Breast Cancer Histopathological Database (BreakHis). Moreover, the dataset comprises microscopic images from 82

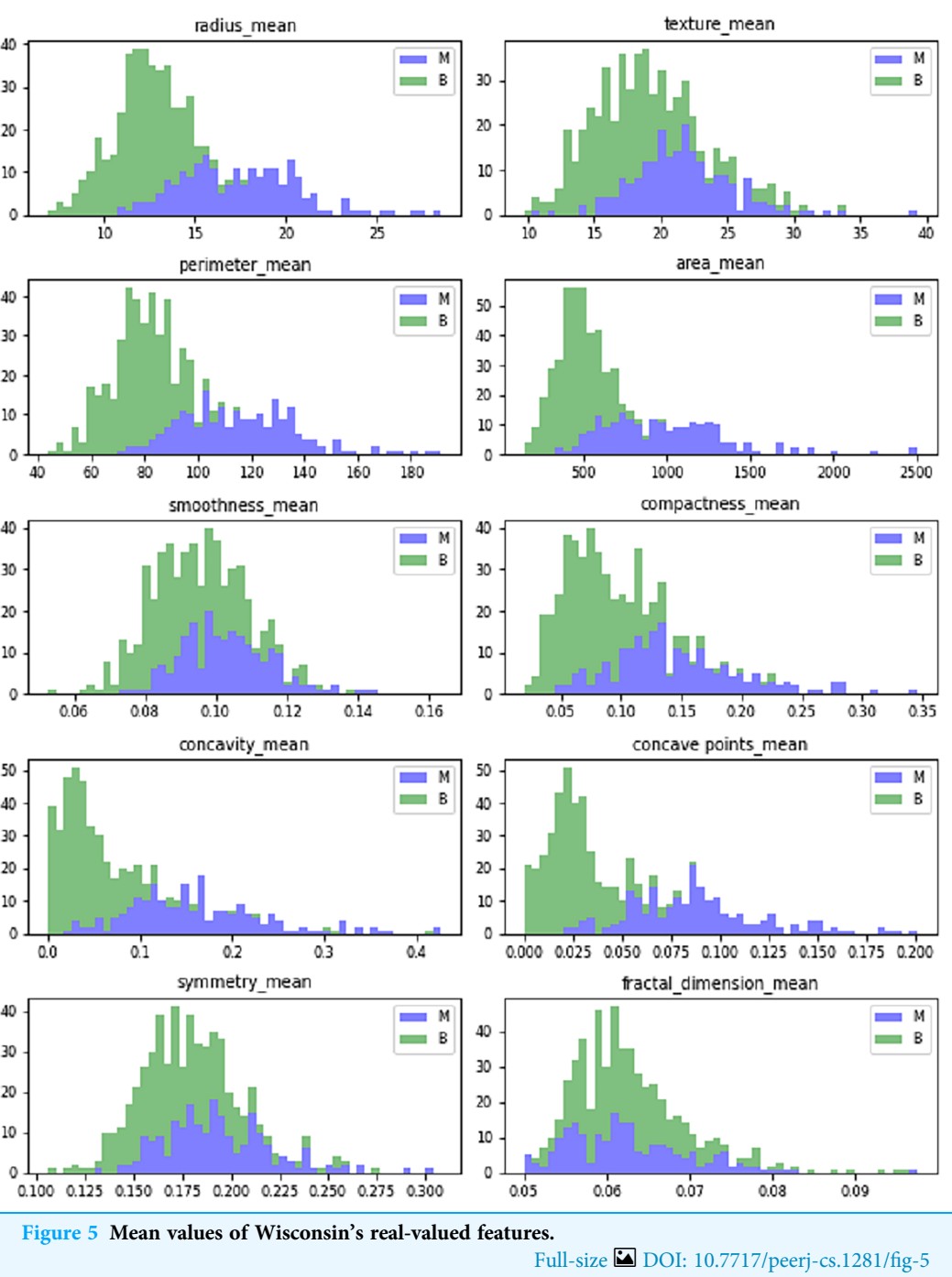

**Figure 5** Mean values of Wisconsin's real-valued features.

different people. Table 4 shows the dataset counts based on the magnification. The magnifying factors are 40×, 100×, 200×, and 400×. The total number of benign in the dataset is 2,480. The total number of malignant data is 5,429.

A large number of data is identified in the 100× magnification. The number of data is 1,437. The least data identified in the magnification is 400×. The amount of data in this field is 1,232.

**Table 3 Feature names of sample data.**

| Parameter | Mean value of sample 1 | Mean value of sample 2 |
| --- | --- | --- |
| Mean radius | 17.99 | 20.57 |
| Mean texture | 10.38 | 17.77 |
| Mean perimeter | 122.8 | 1,32.9 |
| Mean area | 1,001 | 1326 |
| Mean smoothness | 0.1184 | 0.08474 |
| Mean compactness | 0.2776 | 0.07864 |
| Mean concavity | 0.3001 | 0.0869 |
| Mean concave points | 0.1471 | 0.07017 |
| Mean symmetry | 0.2419 | 0.1812 |
| Mean fractal dimension | 0.07871 | 0.05667 |
| Worst texture | 17.33 | 23.41 |
| Worst perimeter | 184.6 | 158.8 |
| Worst area | 2,019 | 1,956 |
| Worst smoothness | 0.1622 | 0.1238 |
| Worst compactness | 0.6656 | 0.1866 |
| Worst concavity | 0.7119 | 0.2416 |
| Worst concave points | 0.2654 | 0.186 |
| Worst symmetry | 0.4601 | 0.275 |
| Worst fractal dimension | 0.1189 | 0.08902 |

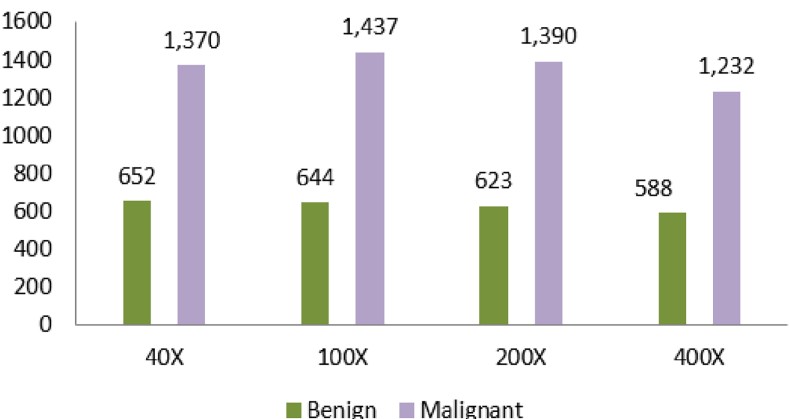

**Figure 6 Different magnification of malignant cancer.**

*Spanhol et al. (2015)* shows the different magnification samples of malignant cancer. The magnificent factor of 400× shows the clear and magnified view of cancer. The enormous size of tissue is removed from the patient under the anesthetic process and the tissue is analyzed to capture the magnified view of the cancer cells. The benign cells are always relatively slow growing and unharmed. Malignant cancer is fast-growing and has the possibility of spreading to the nearest cells and tissues.

Total params: 18,333,506.

**Table 4 Breast cancer histopathological database (BreakHis) dataset counts based on magnification.**

| Magnification | Benign | Malignant | Total |
|---|---|---|---|
| 40× | 652 | 1,370 | 1,995 |
| 100× | 644 | 1,437 | 2,081 |
| 200× | 623 | 1,390 | 2,013 |
| 400× | 588 | 1,232 | 1,820 |
| Total of images | 2,480 | 5,429 | 7,909 |

**Table 5 TLBCM model construction model: Sequential_1.**

| Layer (type) | Output shape | Param # |
|---|---|---|
| densenet201 (Model) | (None, 7, 7, 1920) | 18321984 |
| obal_average_pooling2d_1 | (None, 1920) | 0 |
| dropout_1 (Dropout) | (None, 1920) | 0 |
| batch_normalization_1 (Batch) | 0 | 0 |
| (None, 1920) | 7680 | 0 |
| dense_1 (Dense) | (None, 2) | 3842 |

Trainable params: 18,100,610.

Non-trainable params: 232,896.

Parameter values used in the development of the TLBCM model are shown in Table 5. Dropout layers have become the method of choice in recent years for reducing the overfitting that may occur in neural networks. Batch normalization provides a solution to a significant issue that is known as internal covariate shift. It is beneficial since it makes the data moving between intermediate layers of the neural network appear, and as a result, a faster learning rate is employed. Due to the fact that it has a regularizing impact, dropout may frequently be eliminated.

The total parameters from the input are 18,333,506, and the trainable parameter for the model is 18,100,610. Data augmentation techniques are applied to address the imbalanced distribution of images in subclasses of Breast cancer detection. Therefore, the proposed model uses all the magnification input images separately to extract the features and combine the features to pass through the proposed model.

Table 5 shows the parameter values from the model construction for the DenseNet. The total parameters from the input are 18,333,506, and the trainable parameter for the model is 18,100,610. Data augmentation techniques were applied to address the imbalanced distribution of images in subclasses of Breast cancer detection. The proposed model uses all the magnification input images separately to extract the features and combine the features to pass through the proposed model.

## Hyperparameter tuning

The most widely used optimizer for the medical-related data set to train, and test, is Adam. Compared to SGD, Adam produces good accuracy in recognizing the output. The initial learning rate is taken as 0.0001. For the upcoming epoch, the learning rate is reduced to 0.1 to improve the accuracy of the recognition. The model is trained for the 20 epochs. When the data set comprises a minimum number of medical data, over-fitting will be a serious issue during model testing and training. The dropout layer is added to the model to inactivate some neurons during the training process. The dropout layer is added to the ResNet and then the DenseNet model is with a rate of 0.2. For the fully connected layer, the drop-out value is 0.5.

## RESULT AND DISCUSSION

The results indicated in the article are taken for the breast of this dataset. In the proposed experimental setup, 80% of the data is used for training, and 20% of the data is used for testing. The batch size is set as 64. In each of the iterations, the samples are shuffled to create a new order. The proposed model achieves a higher accuracy, 98.3% compared to the other traditional methods. The maximum of the target output is predicted correctly as malignant and benign. A total of 1.7% of the outputs alone are miss-predicted as malignant instead of benign.

The performance of the model is assessed using the accuracy recall, precision; F1 score, Matthew's correlation coefficient (MCC) and accuracy are the important factors of data collection and measurement. Both precision and accuracy predict the measurement against the original output. Accuracy is the measurement of the predicted value and precision measures the reproducible value even if it deviates from the target output.

$$Accuracy = \frac{Total\ number\ of\ correct\ prediction}{total\ number\ of\ prediction} \tag{6}$$

$$Precision = \frac{True\ positive}{All\ positive} \tag{7}$$

The measurement of True positive recalls and the F1 score is the mean value of precision and recall.

$$F1\ Score = \frac{2 * (Recall * Precision)}{(Recall + Precision)} \tag{8}$$

where,

$$Recall = \frac{correctly\ predicted\ positive\ value}{Total\ positive\ value} \tag{9}$$

Figure 7 shows the receiver operating characteristic curve (ROC) to measure the performance of the proposed classification. The final area under the ROC curve (AUC) is predicted as 0.692. Moreover, the proposed model has a positive and a negative rate. The true positive rate is calculated using the formula.

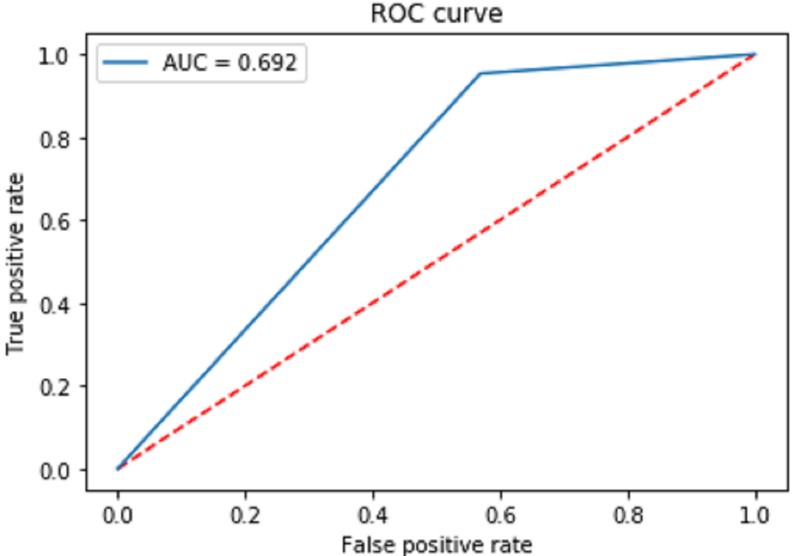

**Figure 7 Receiver operating characteristic curve (ROC) to measure the performance of the proposed classification.**

$$T_r = \frac{T_p}{T_p + F_n} \tag{10}$$

The false positive rate is calculated as:

$$F_r = \frac{F_p}{F_p + T_n} \tag{11}$$

$T_p$ is True Positive and $T_n$ is True negative. $F_p$ is False Positive and $F_n$ is False negative. The AUC Is the aggregate measurement of the performance across all epochs.

## Model evaluation

To evaluate how well the suggested model works, a confusion matrix is calculated. They are represented by a 22 matrix, and they're used to specify which categories they are aiming at. Moreover, the potential victims are divided into two groups; those with no malicious intentions and those who do intend harm. The columns contain the projected values, whereas the rows include the target classes. It is clear from the confusion matrix that the majority of the data samples are correctly identified as benign.

Figure 8 The confusion matrix is used to analyze the performance of the proposed model. It is a 2 × 2 matrix that indicates the target classes. There are two target classes, namely benign and malignant. The predicted values are represented as columns, and the target classes are defined as rows. From the confusion matrix, it is observed that only a few data samples are misclassified as malignant instead of benign. Figure 8 shows the confusion matrix of the TLBCM model, which evaluates the value predicted against the target output. A total of 97.2% of malignant data is predicted correctly, and 99.4% of

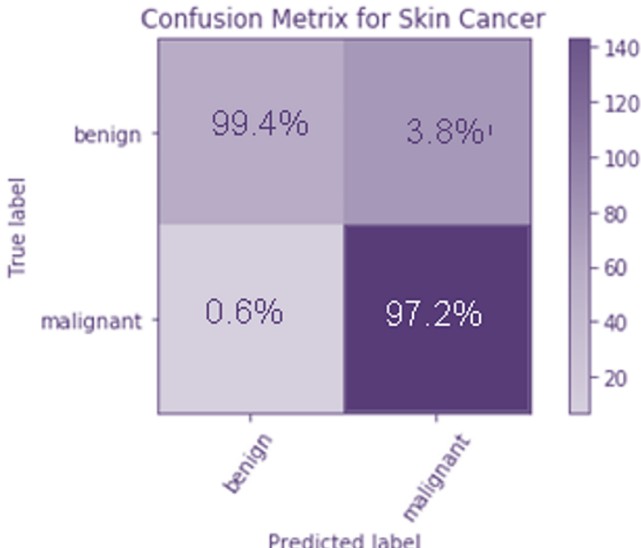

**Figure 8 Confusion matrix of the TLBCM model which evaluates the value predicted against the target output.**

benign data is correctly predicted. A total of 0.6% of data is misclassified as malignant when it is mild. A total of 3.6% of cells are classified as mild when it is malignant.

In addition, the performance of stratified and basic k-fold cross validation in the process of verifying the result e for cancer prediction is explored. In k-fold CV, the dataset is split into k independent folds, of which k−1 folds are used to train the network and the remaining fold is kept aside for testing purposes. k−1 folds are used to train the network. After then, the method is repeated until each fold has been performed once as part of a test set. After then, the ultimate output of the network is calculated by taking the average of the accuracy values received from each test set. The 5-fold cross-validation is used in our proposed model.

The corresponding results obtained from each fold are 93.123, 95.620, 92.632, 98.124, and 97.86. Figure 9 shows loss and accuracy for all iterations of training and testing. The loss value changes continuously from the beginning of the training to the end. At the end of the training, the value of the loss is nearly 0.1. The accuracy of the training starts from 70% and reaches the highest of 98.3%. The sensitivity of the proposed method is 0.99 and the specificity is 0.97. The MCC of our proposed method is 99.2%

Figure 10 shows the mean area against the smoothness of the cancer cell. The values produced after the model improvisation. The values correctly predicted in the orange color values as 1 and the wrongly predicted data was indicated as 0 in the blue color.

## Comparative analysis

The proposed model is compared with several other recent models implemented by other researchers. When compared to the other studies, two types of datasets are used instead of focusing on one dataset which increases the learning capacity of the model. The other added advantage of the proposed work is the pre-trained classifier. In addition to that, the

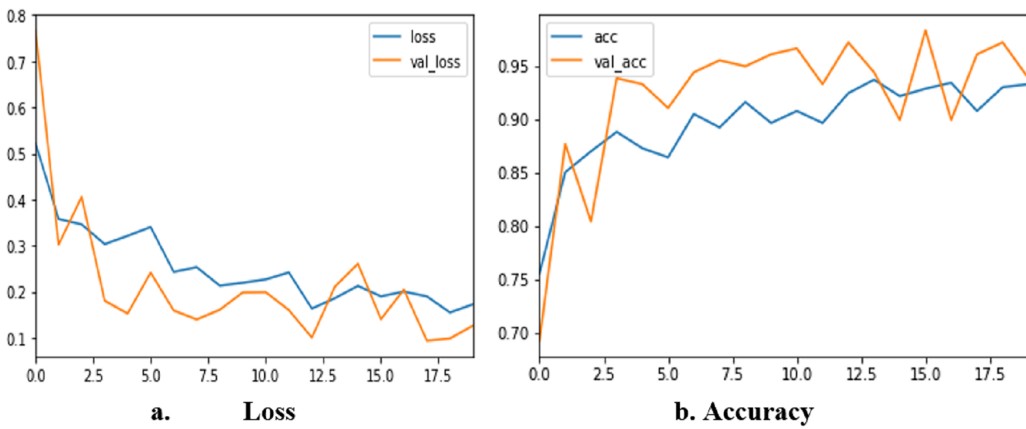

a.    Loss          b. Accuracy

**Figure 9 Accuracy and loss of TLBCM model based on the BreastHis dataset.**

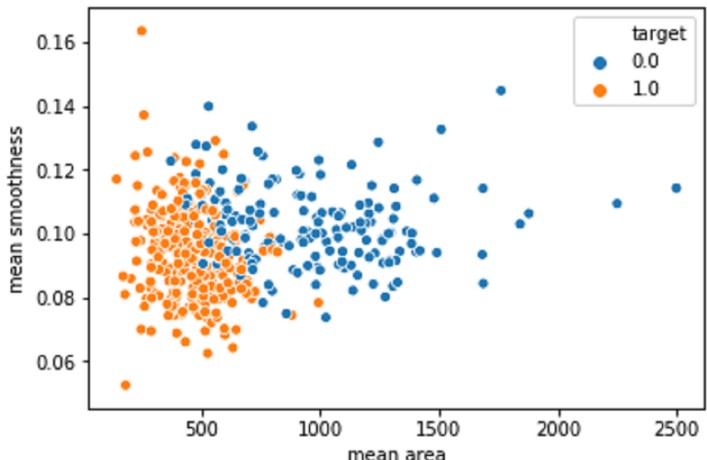

**Figure 10 The mean area against the smoothness of the cancer cell from the BreastHis dataset.**

model has a completely fully connected layer that could be integrated into the pre-trained model as a CNN.

The Table 6 shows the accuracy value of the proposed model against the previous models. The model CNN + LSTM achieves only 82.5% of accuracy and the model ResNet + LSTM achieves 90.1%. The accuracy is increased along with the DenseNet as transfer learning in the proposed TLBCM method.

## CONCLUSION AND DISCUSSION

The machine learning algorithms KNN, SVM, and decision tree are being used by many researchers to detect breast cancer from different data sets. This image feature data is monitored continuously using CPS in the network. Although these methods perform well, the handcrafted extracted features make the model complex, and the detection of cancer is also delayed. The proposed article has chosen the method TLBCM, which combines ResNet, DenseNet, and CNN network for the effective classification of breast cancer to

**Table 6 Proposed model comparison with previous research methods**

| Model | Accuracy | Precision | Recall | F1 score |
|---|---|---|---|---|
| TLBCM | 98.3% | 0.65 | 0.95 | 0.77 |
| CNN + LSTM | 82.5% | 0.62 | 0.85 | 0.70 |
| ResNet + LSTM | 90.1% | 0.59 | 0.92 | 0.75 |
| SVM | 87% | 0.95 | 0.95 | 0.95 |
| Decision tree | 82% | 0.90 | 0.90 | 0.91 |
| ResNet | 93% | 0.92 | 0.93 | 0.95 |
| DenseNet | 97% | 0.96 | 0.95 | 0.94 |

predict whether it is malignant or benign. Furthermore, transfer learning using ResNet and DenseNet is used to determine the characteristics of the dataset. The results of the experiment showed that the transfer of knowledge across unrelated jobs can be beneficial to the target occupation. It is recommended to employ initialization based on previously taught weights to kick off the learning process. Gradually, adjusting the loaded weights is then required to adjust the network to the new dataset. To do this, backpropagation is restarted on the layers with the slowest learning rate. The experimental results show that TLBCM with CPS has performed better compared to the existing method. The dataset Breast Cancer Wisconsin (Diagnostic) and Breast Cancer Histopathological Database (BreakHis) are used in the proposed article. Thus, the proposed work has achieved 98.3% of accuracy on these datasets. The detection of breast cancer through the proposed work helps doctors and pathologists to classify cancer with minimum effort and earlier in nature.

Several suggestions are made about how to optimize the development and education of a neural network model using the given experiments. It is also shown that TLBCM may greatly improve the performance of the model with a little training set. The effectiveness of the most important hyper-parameters is evaluated and compared the results of TLBCM are to the state-of-the-art methods. The effectiveness of TLBCM tactics is also investigated in a range of experimental settings. The consequence is an earlier diagnosis, which means more effective treatment and fewer deaths from breast cancer. A major drawback of the proposed research is the small sample size of histopathology pictures utilized to train the deep-learning models. Nonetheless, *via* the use of transfer learning, the specific issue is being solved. In future research, a variety of deep-learning models is used to achieve patient-level categorization on large histopathology datasets.

Future studies can use a greater variety of datasets for precise breast tissue localization. In order to further describe the semantics label dependence and the significance of the image-label connection, future research will focus on discovering the techniques to make advantage of the unique aspects of the multi-labeled challenge. It is preferable to use other imaging modalities in addition to mammography when training a computer-aided diagnostic system for the early detection of breast cancer. Therefore, it will gain use from additional pre-existing rich representations. Although little blood test data is available for

the proposed study, future investigations will hopefully replicate these findings using considerably bigger datasets of blood test values for a variety of medical illnesses. Additionally, new territories might be broken into in the quest to solve the categorization of prediction issues.

### Funding
The study was supported by the SPEV project, run by the faculty of informatics and management, at the University of Hradec Kralove, Czech Republic. The funders had no role in study design, data collection and analysis, decision to publish, or preparation of the manuscript.

### Grant Disclosures
The following grant information was disclosed by the authors:
University of Hradec Kralove, Czech Republic.

### Competing Interests
The authors declare that they have no competing interests.

### Author Contributions
- Sudha Prathyusha Jakkaladiki conceived and designed the experiments, performed the experiments, analyzed the data, performed the computation work, prepared figures and/or tables, authored or reviewed drafts of the article, and approved the final draft.
- Filip Maly conceived and designed the experiments, performed the experiments, analyzed the data, performed the computation work, prepared figures and/or tables, authored or reviewed drafts of the article, and approved the final draft.

### Data Availability
The data is available at the Breast Cancer Histopathological Database (BreakHis): https://web.inf.ufpr.br/vri/databases/breast-cancer-histopathological-database-breakhis/.
Sudha Prathyusha Jakkaladiki. (2023). An Efficient Transfer Learning Based Cross Model Classification (TLBCM) Technique for the Prediction of Breast Cancer https://doi.org/10.5281/zenodo.7538241.

### Supplemental Information
Supplemental information for this article can be found online at http://dx.doi.org/10.7717/peerj-cs.1281#supplemental-information.

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
