# Peer review of "An efficient transfer learning based cross model classification (TLBCM) technique for the prediction of breast cancer"

_PeerJ Computer Science, doi:10.7717/peerj-cs.1281_

## Round 0.1 · original submission · Major Revisions

Please, revise your paper carefully as per suggestions of the expert reviewers and resubmit.

Reviewer 1 ·

Basic reporting

The article is well written and easy to understand. However, few of my feedback can be considered to improve the quality of the paper but all are not necessary.
1. Introduction may be improved, adding the highlights and the problem statements.
2. Provide the experimental setup and the tools used for the study.
3. If possible provide a simulation parameters table.
4. You could improve writing, link better the ideas flow in the Introduction.
5. Review references because some of them are unstandardized.
6. The conclusion needs improvements towards major claimed contribution.
7. Write some future directions in the conclusion section.
8. The difference between your proposal and related works is not clear, you could do details better. I suggest add a comparative table in ''Related Literature'' to contrast your solution in front of related works.
9. You could discuss the relationship between your solution and past literature.You can add few related recent papers as references: ABCanDroid: a cloud integrated android app for noninvasive early breast cancer detection using transfer learning, Federated learning based Covid‐19 detection etc. These papers used ML/FL in medical fields.

Experimental design

As above

Validity of the findings

As above

Additional comments

As above

Reviewer 2 ·

Basic reporting

I have studied the article you have done in detail.

A paragraph about the organization of the article should be added at the end of the Introduction section. “An Efficient Transfer Learning Based Cross Model Classification (TLBCM) Technique for Breast Cancer
In the study titled “Prediction using Cyber-Physical System”, the authors used resnet and densenet architectures, which are pre-trained models. The study needs a major revision.

In the summary section of the study, the proposed model should be highlighted instead of resnet and densenet architecture.
A paragraph about the developed model should be added to the study.
The model proposed in the results section should be compared with resnet and densenet.
There are performance criteria parameters that are frequently used in the literature. Sensitivity, specficity, False positive rate, false negative rate etc.
In the Discussion section, the study should be compared with similar studies in the literature in a table.
Limitations of the study should be given clearly.
Figure 5,7,13 should be removed.
Spelling errors in the study should be corrected. For example “The table 5 Shows the accuracy value…”,” Figure ?? The confusion matrix is ​​used to analyze the performance”.
It is sufficient to give the abbreviations in the first place. It does not need to be given repeatedly.

Experimental design

The proposed method in the study should be redesigned to bring to the fore. For the proposed model, a subtitle and an explanatory figure would be more appropriate.

Validity of the findings

Densenet and resnet alone are not enough. The proposed model should be in the foreground.

Additional comments

Updating the study in line with the specified revision will increase the quality of the study.

Reviewer 3 ·

Basic reporting

1. The manuscrip is poorly written: redundant, some important details are missing, or sentence doesn't make sense, standard terminologies wrongly written. Language edition with native speaker is mandatory. The entire manuscript from Abstract to Conclusion and discussion must be changed.
2. Important references regarding transfer learning in breast cancer are not provided. For example, https://doi.org/10.3390/cancers13040738 and https://doi.org/10.1016/j.compeleceng.2022.108468
3. The manuscript must include what is the challenge, what has been done to overcome that, what is the gap, and finally, what is proposed.

Experimental design

1. There is no rationale of why the proposed method is used and/or, there is no specific justification of what is being proposed in the manuscript.
2. The proposed method is not described. Paragraph about TLBCM is given only in lines 168-171, which is not even clear. Authors must describe in detail what their proposed method is.
3. The two datasets used are not related and there is no justification of why they were used. Wisconsin dataset has different features, not only images, and no detail is given what has been done here. The BreakHis dataset has four resolutions and no detail about what resolution is used in this manuscript.
4. Parameters of deep learning used are not justified. There is no explanation of why these parameter values are used.
5. Regarding comparative analysis done, there is no detail about what control measures have been used? Are the parameters set the same? Are the architecture of transfer learning the same? Why not other CNN models? Why didn't you use recent deep learning models (CNNs, vision transformers)?
6. Experimental design would be improved if the same setting is used and state-of-the-art architectures are used for comparison.

Validity of the findings

1. Figures are not self-explanatory. Nothing written in the text says anything about the figures.
2. Figures should be described well in footnote. The result figures are ambigious; For which dataset are the output figures? Detail should be given.
3. The results are from a single run, I guess. The results should be given in terms of cross-validation (5 fold or others). Since different results are obtained for multiple run, there should be some way to make the results smooth.

Additional comments

1. The title and the work reported has no relation. Consider changing the title with a proper title that describes the work done.
2. Provide detail about the proposed method.
3. Use more statistical measures such as kappa or Matthew's correlation coefficient for comparing against state-of-the-art methods (in the past two years, 2020-2022)

---

## Round 0.2 · accepted · Accept

Dear Authors,

Thank you for the revision. Your paper was accepted following peer review process.

Reviewer 2 ·

Basic reporting

The authors have comprehensively addressed the missing parts of the study in the revision.

Experimental design

The authors have comprehensively addressed the missing parts of the study in the revision.

Validity of the findings

The authors have comprehensively addressed the missing parts of the study in the revision.

Additional comments

The authors have comprehensively addressed the missing parts of the study in the revision.

Reviewer 3 ·

Basic reporting

The manuscript is much better now.

Experimental design

The manuscript is much better now.

Validity of the findings

The manuscript is much better now.

Additional comments

The manuscript is much better now.